# Low-dose aspirin is not effective as an adjunct treatment for HIV infection among people living with HIV on dolutegravir-based antiretroviral therapy: A randomised double-blind, parallel-group placebo-controlled trial

Tosi M. Mwakyandile[1]*, Grace A. Shayo[2], Philip G. Sasi[1], Peter P. Kunambi[1], Ferdinand M. Mugusi[2], Godfrey Barabona[3], Takamasa Ueno[3,4], Eligius F. Lyamuya[4,5]

1 Department of Clinical Pharmacology, School of Biomedical Sciences, Campus College of Medicine, Muhimbili University of Health and Allied Sciences, Dar es Salaam, Tanzania, 2 Department of Internal Medicine, School of Clinical Medicine, Campus College of Medicine, Muhimbili University of Health and Allied Sciences, Dar es Salaam, Tanzania, 3 Division of Infection and Immunity, Joint Research Centre for Human Retrovirus Infection, Kumamoto University, Kumamoto, Japan, 4 Department of Microbiology and Immunology, School of Diagnostic Medicine, Campus College of Medicine, Muhimbili University of Health and Allied Sciences, Dar es Salaam, Tanzania, 5 Collaboration Unit for Infection, Joint Research Centre for Human Retrovirus Infection, Kumamoto University, Kumamoto, Japan

* tosimwakys@gmail.com, tosi.mwakyandile@muhas.ac.tz

OPEN ACCESS

## Abstract

### Background

Despite virologic suppression with antiretroviral therapy (ART), immune activation (IA) in people living with HIV (PLHIV) remains high and is linked to non-AIDS complications. Alongside its other virologic and immunologic benefits, aspirin promisingly appears to lower the residual IA in PLHIV in small studies.

### Methods

We conducted a double-blind, parallel-group randomised trial involving ART-naïve PLHIV initiating ART at recruitment. Participants were randomly assigned (1:1) to receive 75 mg aspirin or placebo daily for 24 weeks, alongside standard of care. The primary outcome was proportion of participants attaining HIV viral load < 50 RNA copies/mL at weeks 8, 12 and 24. Secondary outcomes assessed at 12 and 24 weeks were CD4 count, platelet and monocyte activation (soluble P-selectin and soluble CD14, respectively), T-cell activation (CD69 expression, CD38/HLA-DR co-expression) and T-cell exhaustion (PD-1 expression). Data were analysed by intention-to-treat strategy. Between-treatment arm comparisons were made by regression models using generalised estimating equations. Competing risk analyses were employed for morbidity, all-cause mortality and adverse events (AE).

**Data availability statement:** All relevant data are within the manuscript and its Supporting Information files.

**Funding:** The trial was funded by the Transforming Health Education in Tanzania (THET) project and the HIV Implementation Science Training Grant of Fogarty International Centre (FIC) of the National Institutes of Health (NIH) (Award Numbers 1R25TW011227-01and 5D43 TW009775-03, respectively). Additional funding was provided by the Swedish International Development Cooperation Agency (Sida), the Japan Society of Promotion of Sciences (JSPS), namely, JSPS KAKENHI Grant-in-Aid for Scientific Research 21K19657, 22H03119, 22KK0148, JSPS Bilateral Open Partnership Joint Research Project, JPJSBP120219933 and JPJSBP120239932, JSPS Core-to-Core Program, JPJSCCB20190009 and JPJSCCB20220010 and the Japan Student Service Organisation (JASSO). The content is solely the responsibility of the authors and does not necessarily represent the official views of the NIH and the other funders.

**Competing interests:** The authors have declared that no competing interests exist.

## Results

Out of 430 recruited participants, 216 were randomised to aspirin and 214 to placebo arms, with 112 and 131 participants completing the study, respectively. Proportions of participants attaining primary outcome at week 24 were comparable (78.4% aspirin arm versus 80.67% placebo arm, p = 0.53). There was larger decrease in CD8$^+$CD69$^+$ (%) at 12 weeks only in the placebo arm (median change (IQR): −0.42 (−2.07, 0.33) versus −0.06 (−1.30, 0.90) in the aspirin arm, p = 0.04). Other markers and secondary outcomes: morbidity (35.6% versus 34.5%), mortality (4.63 versus 3.27 per 100 person-weeks) and AEs (96.7% versus 98.0%) were similar between the aspirin and placebo arms, respectively, p > 0.05.

## Conclusions

Low-dose aspirin initiated alongside ART through 24 weeks did not impact virologic or immunologic markers among PLHIV.

## Trial registration

PACTR202003522049711, NCT05525156

## Introduction

People living with HIV (PLHIV) currently experience improved life expectancy, attributed to a decrease in AIDS-related deaths [1,2]. Antiretroviral therapy (ART), known to reduce HIV viral load (HVL) and increase CD4 count, has played a key role in improving survival [2]. However, now PLHIV face non-AIDS complications such as cardiovascular diseases [3], linked to immune activation (IA) and inflammation [4,5]. Chronic IA and inflammation are higher in PLHIV than in the general population [6–9], and their causes are multifactorial, including HIV itself, ART, microbial translocation, and co-infections [10,11]. Notably, activated platelets, generally higher in PLHIV, are thought to contribute to IA [7,11]. Although virologic suppression resulting from ART is associated with a significant reduction in IA and inflammation, it does not restore the levels to an HIV-uninfected state [12–15]. The effect of ART on platelet activation (PA) is debated; with studies reporting reduction or normalisation [15,16] while others report no effect [16,17]. Additionally, some ART regimens, such as protease inhibitors and abacavir (ABC)-based regimens, have been associated with platelet hyperactivity [16,18–20]. These limitations of ART to effectively correct the IA and/or inflammation suggest a need to explore additional approaches to address the limitations.

Early ART initiation after diagnosis is currently practised but is not fully successful in controlling IA or inflammation [21–24]. Several reasons are thought to explain this failure, including the timing of ART initiation since HIV infection diagnosis, duration of ART, mode of acquiring HIV and presence of co-morbidities such as hepatitis C infection [21]. Furthermore, early ART initiation suffers diagnostic, operational, behavioural and socioeconomic hindrances [25]. Potentially, an add-on drug to ART, preferably

with antiplatelet and/or anti-inflammatory properties, can be employed in addition to early ART initiation to normalise the deranged immune system, hoping to decrease the prevalence of non-AIDS complications and their associated morbidity and mortality.

Several candidate drugs have been explored in clinical trials [10]. For example, statins, angiotensin-converting enzyme inhibitors, angiotensin II receptor blockers and dipeptidyl peptidase 4 inhibitors have shown promising results in preliminary studies, some prompting larger studies [26–31]. These drugs, however, are costly, and should they be approved for routine care of PLHIV, their adoption may be challenging, especially in low and middle-income countries like Tanzania.

Acetylsalicylic acid (ASA), also known as aspirin, is a cheaper, commonly prescribed drug licensed as an antiplatelet in low doses as well as an analgaesic and anti-inflammatory in moderate to high doses. Low-dose aspirin is also indicated in the general population for the prevention of myocardial infarction and preeclampsia.

In PLHIV, aspirin has yielded mixed results concerning IA and PA [7,32]. In two small trials in the USA among ART-virologically suppressed PLHIV, a week of low-dose aspirin dampened IA and PA in one, while in the other, neither 100 mg nor 300 mg aspirin for 12 weeks dampened IA markers [7,32]. To clarify these contradictory findings, relatively larger and longer trials are needed. Additionally, two studies, both conducted in Zimbabwe among ART-naïve PLHIV, demonstrated virologic benefits of the use of aspirin [33,34]. In the first 12-month, three-arm, placebo-controlled pilot study, a daily dose of four times 300 mg aspirin alone or with chloroquine prevented an increase in HVL, whereas the placebo arm showed an increase [33]. In the second study, combining 300 mg aspirin taken four times daily with chloroquine and micronutrients over six months led to a decrease in HVL, which decreased further during an additional three months of aspirin and micronutrients only without chloroquine [34]. These studies and one other also reported that the administration of aspirin resulted in an increase in CD4 counts among ART-naïve PLHIV [33–35]. However, in these studies, aspirin was administered with other drugs and/or micronutrients, making it challenging to attribute the observed effects solely to aspirin. Furthermore, very high doses of aspirin, known to cause serious adverse effects irrespective of duration of use [33–35], were used in these studies. Incorporating aspirin in such high doses in the routine care of PLHIV will not be safe. Besides, it is argued that IA in PLHIV is attributed to PA [7,11], making low-dose aspirin more suitable due to its antiplatelet effects. None of the previous studies were designed to investigate the effects of aspirin on both virologic and IA outcomes [7,32–35]. Therefore, in a larger single-centre phase IIA double-blind, parallel-group randomised controlled trial, we have examined the effects of low-dose (75 mg) aspirin over 24 weeks on HVL, CD4 count, PA, IA and exhaustion among ART-naïve PLHIV initiating ART. We hypothesised that low-dose aspirin in combination with ART in treatment-naïve PLHIV initiating ART would be associated with earlier and sustained virologic suppression and better immunological and clinical responses than ART alone.

## Materials and methods

### Study design and settings

This was a randomised double-blind parallel-group placebo-controlled phase IIA clinical trial. The trial was conducted at HIV care and treatment centres (CTCs) in three different hospitals (Mbagala Rangi Tatu Hospital; Mwananyamala and Temeke regional referral hospitals) in Dar es Salaam, Tanzania, between 02 March 2020 and 22 June 2022. The trial was terminated due to the expiry of the study drugs. In addition to standard ART, participants were randomised to receive a daily dose of 75 mg aspirin or placebo for 24 weeks. All the CTCs are managed by a non-governmental organisation, Management and Development for Health (MDH), in coordination with the Tanzanian government through the National AIDS, STIs and Hepatitis Control Programme.

### Study population and participants' selection

Consenting ART-naïve PLHIV, either newly recruited or diagnosed, aged 18 years or older, initiating ART, willing to attend their respective CTCs and stay in Dar es Salaam for the duration of the trial, were enrolled by the study doctor. Those

with a history of intolerance to aspirin or aspirin-containing products, asthma, a history of, or active peptic ulcer disease, severe renal disease (estimated glomerular filtration rate- eGFR < 30 mL/min/1.73 m$^2$), as well as those who were predisposed to bleeding, on antithrombotic or trial-prohibited drugs (S1 File) and pregnant women were excluded.

## Randomisation and blinding

A randomisation list was computer-generated by a biostatistician not directly involved in the trial. A 1:1 randomisation ratio in blocks of 10 was implemented. The list was sent to a non-blinded pharmaceutical technician at the Muhimbili University of Health and Allied Sciences (MUHAS), who stored the drugs as per manufacturer's recommendation (below 30°C). Study drugs for each participant were packed in boxes, labelled by applying a study number on them and sent to study sites monthly for dispensing by a blinded study pharmacist. All other study staff and the participants were blinded. The randomisation code was made available to the investigator after approval by the data and safety monitoring board (DSMB), once all data were collected and analysed.

## Intervention

The active study drug was a blister-packaged enteric-coated 75 mg aspirin tablet, whereas the placebo was similar to the active drug in terms of packaging, colour, size and shape.

In addition to their ART regimen, participants swallowed an active drug (Cardisprin 75, Cosmos, Nairobi, Kenya) or placebo from the same manufacturer with clean drinking water every evening postprandial for 24 weeks.

All participants received ART according to the Tanzanian standard of care for PLHIV. The default ART regimen was a fixed-dose combination of Tenofovir Disoproxil Fumarate (TDF)+ Lamivudine (3TC) + Dolutegravir (DTG) [36]. A fixed-dose combination of ABC + 3TC + DTG or TDF + 3TC + Efavirenz (EFV), as well as Zidovudine (AZT) + 3TC + DTG, was initiated as the default regimen in special situations such as intolerance or contraindications to other regimens. Participants who were currently treated for tuberculosis received a double dose of DTG. Cotrimoxazole prophylactic treatment was initiated if the CD4 count was ≤ 350 cells/µL at ART initiation, continuing until the CD4 count exceeded 350 cells/µL or virologic suppression was achieved, provided there were no contraindications. Participants presenting with a CD4 count < 200 cells/µL and a positive cryptococcal antigen screening test at ART initiation received pre-emptive therapy with fluconazole 800 mg once a day for up to two weeks. This was followed by 400 mg of fluconazole daily for eight weeks, then 200 mg, once daily. Therapy was stopped when the CD4 count was ≥ 100 cells/µL with virologic suppression or the CD4 count was ≥ 200 cells/µL. Additionally, two to four weeks after ART initiation, all participants not currently treated for tuberculosis received Tuberculosis Preventive Therapy with isoniazid daily for at least six months after ruling out active tuberculosis [36].

## Patient follow-up and evaluation

Sociodemographic information was collected from each participant during enrolment, and also blood samples for HVL, CD4 count, monocyte (soluble CD14) and platelet (soluble P-selectin) activation, as well as T-cell activation (CD4$^+$ and CD8$^+$ singly expressing CD69 and co-expressing CD38 and HLA-DR), and T-cell exhaustion (PD-1). Each participant was seen biweekly for the first month and thereafter monthly for six consecutive months. At each monthly follow-up, participants underwent a thorough medical history and physical examination to ascertain if they developed adverse events (AEs), visited or were admitted to a health facility for medical attention/care due to illness. All AEs were documented and followed up until their resolution or the end of the study. Those deemed serious were reported to the DSMB and the MUHAS and national ethics committees within 72 hours of their awareness by the investigator. Before drug refills, monthly pill counting of participants' leftover tablets was done for both study drugs and ART to assess adherence. Adherence was calculated by subtracting the pills remaining in a particular month from the total pills dispensed in the month prior (N). The

difference was divided by N and multiplied by 100 to determine monthly percentage adherence. The overall adherence was determined by averaging the monthly adherence. For participants who did not complete the entire study, their overall adherence was calculated based on their average for the months they participated in the study. Adherence to study drugs and ART was considered good if overall percentages were ≥ 71% and > 90%, respectively.

## Laboratory procedures

The details of these procedures have been reported elsewhere [37]. In brief, each participant provided venous blood at baseline, weeks eight, 12 and 24. At baseline and week 24, we collected venous blood in K2 EDTA vacutainer tubes in aliquots of 4 mL for CD4 count and full blood picture (FBP) analysis and 12 mL for HVL, sCD14, sP-selectin, peripheral blood mononuclear cells - PBMCs separation for markers of T-cell activation and exhaustion, and another 4 mL in red topped vacutainer tube for renal and liver function tests. At week 12, we collected venous blood in K2 EDTA vacutainer tubes in aliquots of 2 mL for CD4 count analysis and 12 mL for HVL, sCD14, sP-selectin, and PBMCs separation for markers of T-cell activation and exhaustion. These were transported in cool boxes from sites to laboratories situated at MUHAS within two hours of collection. FBP analysis using Sysmex analyser (Sysmex Corporation, Japan) and CD4 count using FAC-SPresto (BD Biosciences, San Jose, California, USA) were done within 24 hours of sample collection. Within six hours of phlebotomy, the aliquot in the red-topped vacutainer tube was centrifuged to obtain serum, which was analysed for serum creatinine, alanine aminotransferase (ALT) and aspartate aminotransferase (AST) using COBAS Integra 400 Plus, Roche Instruments Centre AG, Rotkreuz, Switzerland. PBMCs were freshly isolated from a 12 mL aliquot of whole blood using Ficoll-Paque Plus media solution (GE Healthcare Life Sciences Inc., Chicago, Illinois) through density gradient centrifugation. The isolated PBMCs were stored in liquid nitrogen until shipment and analysis at the end of the study.

The resultant plasma was stored in 1.5 mL and 4.5 mL aliquots at minus eighty (−80)°C. The 1.5 mL portion was later used for HVL testing (COBAS AmpliPrep/COBAS Taqman HIV-1 quantitative test, Roche Diagnostics, Switzerland) performed at the Muhimbili National Hospital's Central Pathology Laboratory in Tanzania. The 4.5 mL portion and the PBMCs were shipped in dry ice to Japan to analyse soluble and cellular markers of PA and IA and exhaustion, respectively.

## T-cell activation and exhaustion

After thawing, PBMCs were labeled with 1:100 diluted anti-CD3 FITC, anti-CD4 BV510, anti-CD8 APCcy7, anti-CD14 PerCP, anti-CD19 PerCP, anti-CD69 BV421, anti-CD38 PE, anti-HLA-DR APC, and anti-PD-1 PEcy7. One per cent formaldehyde-fixed cells were acquired on BD FACSCanto™ II (BD Biosciences, San Jose, California, USA) and analysed with FACSDiva software (BD Biosciences, San Jose, California, USA). The results were obtained on FlowJo™ version 10.8.2 software (TreeStar, Ashland, Oregon) and expressed as percentages of CD69+ (for acute activation), CD38+HLA-DR+ (for chronic activation), and PD-1+ (for exhaustion) among CD4+ and CD8+ T lymphocytes.

**Soluble CD14 (sCD14) and soluble P-selectin (sP-selectin) measurement.** Levels of sCD14 and sP-selectin were measured in 1:1000 and 1:50 diluted plasma, respectively, by Cytometric Beads Array kit (BD Biosciences, San Jose, California, USA) according to the manufacturer's manual. FACSCanto™ II (BD Biosciences, San Jose, California, USA) was used to acquire the samples, data analyses were performed using FCAP array software (Soft Flow Hungary Ltd., Hungary), and levels of the markers were expressed as picograms per millilitre (pg/mL).

## Study outcomes

The trial's primary outcome was the proportion of study participants attaining HVL < 50 RNA copies/mL (virologic suppression) at weeks eight, 12 and 24. Unfortunately, in contrast to initial plans, we did not have the resources to test the HVL at eight and 12 weeks. Secondary outcomes included the proportion of participants with HVL ≥ 1000 RNA copies/mL (virologic failure) at 24 weeks, the proportion with a > 30% increase in CD4 count from baseline value at 12 weeks, changes

from baseline in CD4 count, and decreases in markers of PA, IA, and exhaustion over 24 weeks. Other secondary outcomes included the rate of morbidity (a visit or admission to a health facility for medical attention/care due to illness), all-cause mortality, and AEs over 24 weeks. For abnormal laboratory values, anaemia was defined as a haemoglobin concentration of < 12 g/dl for females and <13g/dl for males. Thrombocytopenia was defined as a platelet count of < 150 x 10$^3$ cells/μL. Elevated ALT and AST were defined as ≥ 1.25 x upper limit normal (ULN) values. For ALT, the ULN for females was 34 IU/L and 45 IU/L for males, and for AST, the ULN were 31 IU/L and 35 IU/L for females and males, respectively. Chronic kidney disease (CKD) was defined as eGFR < 90 mL/min/1.73 m$^2$.

## Study sample size and statistical analysis

With 75% of patients on a DTG-based ART alone achieving HVL < 50 RNA copies/mL at eight weeks [38], and assuming that adding aspirin to ART improves this proportion by 15%, each treatment arm required 227 participants to achieve 80% statistical power, with a 10% expected loss to follow-up and a 95% confidence interval.

Data was analysed by intention-to-treat strategy. Categorical variables were presented as frequencies and percentages. Numerical variables were represented by median and interquartile range (IQR). The distribution of baseline characteristics of participants in the treatment arms was compared using the chi-square test for categorical variables or the Mann-Whitney U test for numerical variables as appropriate.

A logistic regression model using generalised estimating equations (GEE) was applied to assess the differences between the treatment arms in proportions of participants with virologic suppression or failure at baseline and 24 weeks. The model had the treatment arm as the only predictor. McNemar test was used to compare the proportions of participants in the treatment arms with HVL < 50 and HVL ≥ 1,000 RNA copies/mL at baseline and 24 weeks. Linear regression model using GEE was used to analyse differences in CD4 count, markers of PA, IA, and exhaustion between the arms at baseline, 12 and 24 weeks. In both types of analyses, the interaction of time and treatment arm represented the treatment effect.

Competing risk analyses were employed for morbidity (mortality and loss to follow-up as competing risks) and mortality (loss to follow-up as competing risk). Median changes in haemoglobin concentration, platelet count, AST and ALT levels and eGFR were calculated and compared between treatment arms using the Mann-Whitney U test. Proportions of participants with abnormal laboratory parameters at week 24 were determined and compared by the Fisher Exact test between treatment arms. All statistical analyses were done using IBM SPSS Statistics Windows version 26 (IBM Corp., Armonk, NY, USA). Competing risk analyses were done in R programming language (in R-studio Desktop version 4.2.1 as integrated development environment) using cmprsk, ggplot2, survival and survminer R packages. A p-value of < 0.05 was considered statistically significant.

## Ethical approval and trial registration

Ethical approval for this trial study was obtained from the MUHAS Research and Ethics Committee (DA.282/298/01/C) and the National Health Research Ethics Committee-NatHREC (NIMR/HQ/R.8a/Vol. IX/3001). The protocol and the study drugs were also approved by the Tanzania Medicine and Medical Devices Authority (TFDA0019/CTR/0003/03, authorisation number TZ19CT0008). Permissions to conduct the trial were obtained from the participating hospitals' administrations. Participants were assigned a study-specific number to achieve confidentiality instead of using their names. All participants provided written informed consent before enrolment, and for those who could not read and write, informed consent was taken via thumbprint with a treatment supporter as a witness. The trial adhered to ICH GCP guidelines and the Declaration of Helsinki (Version 2013). A clinical trial participation insurance was purchased for all participants. The trial was registered before the commencement of recruitment (PACTR202003522049711) and later (NCT05525156).

Participants reporting AEs were managed accordingly or referred for further evaluation and management. Those with AEs deemed related to aspirin were withdrawn from the study treatment but continued receiving standard of care and attending scheduled visits, whenever possible, for AE outcome monitoring.

## Data safety and monitoring

A DSMB, comprising a physician, a biostatistician, and a clinical pharmacologist, was established to oversee the safety of trial participants. Safety assessments were conducted at 43%, 77%, and 83% of the intended recruitment.

## Results

A total of 681 PLHIV were screened for the study, of whom 430 were eligible and randomised. One hundred and forty-six subjects were seen at 12 weeks, and 112 at 24 weeks in the aspirin arm, and 157 at 12 weeks and 131 at 24 weeks in the placebo arm (Fig 1). The high loss to follow-up was related to the COVID-19 pandemic, which caused an initial suspension of recruitment, longer inter-visit intervals and the loss of some patients from CTCs. Adherence to study drugs was good in 97.9% of participants in the aspirin arm and 97.4% of participants in the placebo arm, p = 1.00). Adherence to ART was good in 86.8% and 90.2% of participants in aspirin and placebo arms, respectively p = 0.30).

### Baseline socio-demographic and clinical characteristics

All socio-demographic and clinical characteristics were comparable between the two treatment arms except for the median expression of CD8$^+$CD69$^+$ T cells, which was higher among participants in the placebo arm 2.78 (1.47, 5.03) than in the aspirin arm 2.33 (1.10, 4.09), p = 0.04. The median (IQR) age of all the participants was 37 (28, 45) years, and females were the majority, 279 (64.9%). (Table 1).

### Effect of aspirin on virologic response

While over three-quarters of participants whose HVL were measured at week 24 attained virologic suppression, there was no significant difference between the treatment arms on the virologic response (p = 0.53). For virologic failure, the interaction of treatment arm and time had p = 0.95, Table 2 and S1 Table.

### Effect of aspirin on immunological response

The proportion of participants attaining a > 30% increment in CD4 count at week 12 from baseline in the aspirin arm 92/140 (65.71%) was not significantly different from that in the placebo arm 103/146 (70.55%), p = 0.38.

We observed a significantly larger decrease in CD8$^+$CD69$^+$ % at week 12 (median change (IQR): −0.42 (−2.07, 0.33) in the placebo arm versus −0.06 (−1.30, 0.90), p = 0.04) in the aspirin arm but not at week 24 (median change (IQR): −0.30 (−1.77, 0.76) placebo arm versus −0.05 (−1.37, 0.80), p = 0.53) aspirin arm. Median changes in all other markers did not differ between the treatment arms (p > 0.05). CD4 counts significantly increased while CD4$^+$HLA-DR$^+$CD38$^+$ %, CD8$^+$PD-1$^+$ % and CD8$^+$HLA-DR$^+$CD38$^+$ % significantly decreased from baseline to week 12 and week 24 (p < 0.001), Table 3, S2 Table and S1-S3 Figs.

### Effect of aspirin on clinical response

Among participants who made at least one follow-up visit since enrolment, morbidity was reported by 69/194 (35.6%) of participants in the aspirin arm and 69/200 (34.5%) in the placebo arm (p = 0.82). There were no significant differences in cumulative incidence of morbidity among participants in the aspirin arm (31.94 per 100 person-weeks) and the placebo arm (32.24 per 100 person-weeks), p = 0.97, Fig 2.

### Assessment of safety and tolerability outcomes

Six participants in the aspirin arm withdrew from the study because of AEs related to aspirin (epigastric pain/ peptic ulcer disease in two participants, intolerance in two participants, asthmatic attacks in two participants) compared with five participants (epigastric pain/ peptic ulcer disease in four participants, signs of bleeding in one participant), in the placebo arm.

 

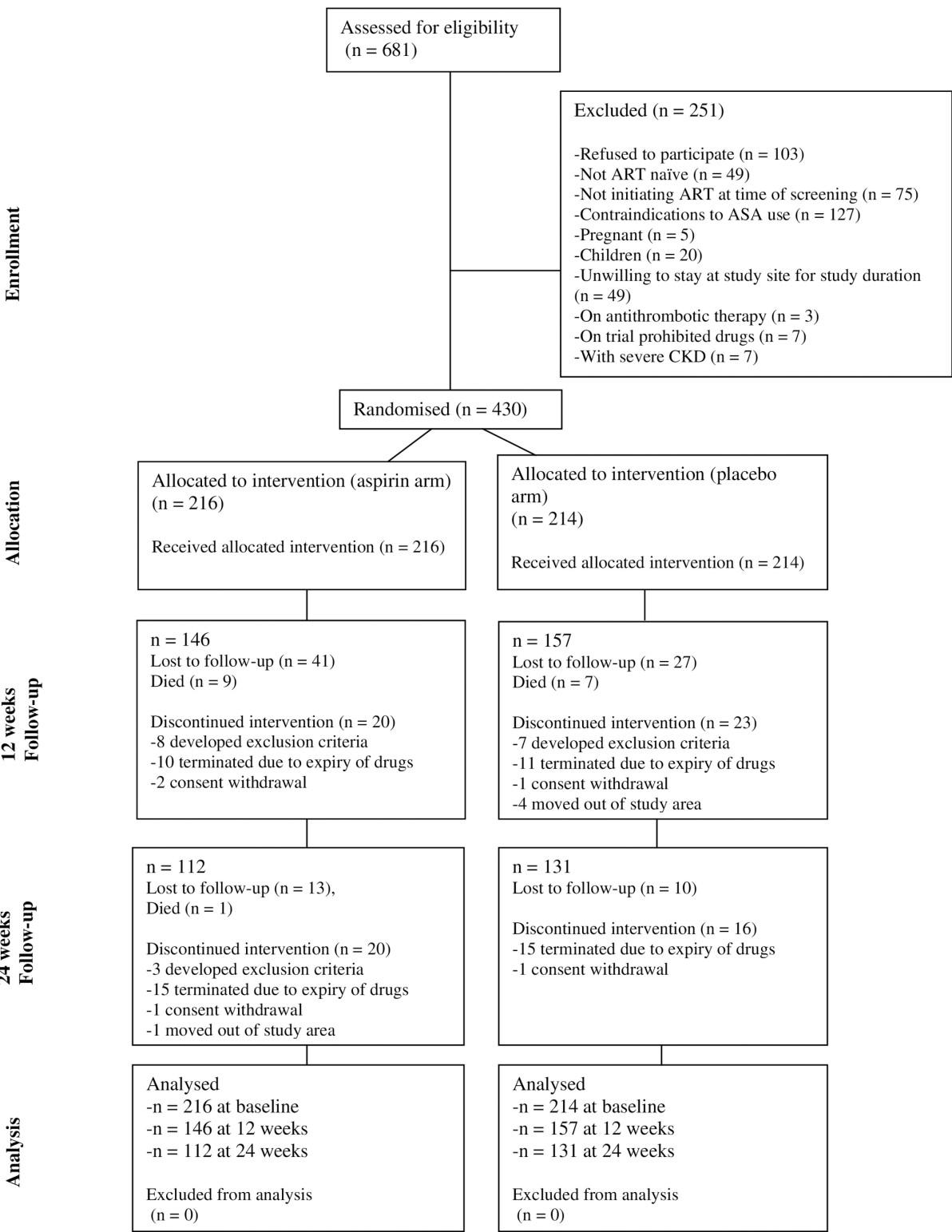

**Fig 1. Consolidated Standards of Reporting Trials flow diagram for participants in the trial.**

**Table 1. Socio-demographic and clinical characteristics of HIV-infected treatment-naïve individuals initiating ART.**

| Variable | | Aspirin arm n=216 | Placebo arm n=214 | p- value* |
|---|---|---|---|---|
| Sex n (%) | | | | |
| | Female | 142 (65.7) | 137 (64.0) | 0.71 |
| Age (years) median (IQR) | | 37 (28,45) | 36 (28,45) | 0.94 |
| Level of education n (%) | | | | |
| | Informal | 18 (8.3) | 16 (7.5) | 0.71 |
| | Primary level | 121 (56.0) | 129 (60.3) | |
| | Secondary level | 62 (28.7) | 52 (24.3) | |
| | University or college | 15 (6.9) | 17 (7.9) | |
| Employment status n (%) (216 aspirin arm, 213 placebo arm) | | | | |
| | Employed | 44 (20.4) | 34 (16.0) | 0.48 |
| | Self employed | 132 (61.1) | 135 (63.4) | |
| | Unemployed | 40 (18.5) | 44 (20.7) | |
| Marital status n (%) (214 aspirin arm, 214 placebo arm) | | | | |
| | Single | 49 (22.9) | 62 (29.0) | 0.20 |
| | Married/Cohabiting | 101 (47.2) | 84 (39.3) | |
| | Divorced/Widowed | 64 (29.9) | 68 (31.8) | |
| Residence n (%) | | | | |
| | Dar es Salaam | 197 (91.2) | 205 (95.8) | 0.05 |
| | Out of Dar es Salaam | 19 (8.8) | 9 (4.2) | |
| CD4 count (cells/mm$^3$) median (IQR), (214 aspirin arm, 210 placebo arm) | | 271.50 (108.75, 471.25) | 286.50 (107.25, 543.00) | 0.81 |
| CD4 count (cells/mm$^3$), n (%) | | | | |
| | < 200 | 87 (40.7) | 84 (40.0) | 0.81 |
| | 200-350 | 43 (20.1) | 38 (18.1) | |
| | > 350 | 84 (39.3) | 88 (41.9) | |
| HIV load (RNA copies/mL) median (IQR), (172 aspirin arm, 174 placebo arm) | | 81650.00 (4935.00, 281385.75) | 51031.00 (6011.50, 246882.25) | 0.56 |
| HIV load (RNA copies/mL) n (%) | | | | |
| | < 50 | 22 (12.8) | 20 (11.5) | 0.83 |
| | 50-999 | 6 (3.5) | 8 (4.6) | |
| | ≥ 1000 | 144 (83.7) | 146 (83.9) | |
| eGFR (mL/min/1.73m$^2$) median (IQR), (198 aspirin arm, 198 placebo arm) | | 109.42 (85.50, 142.06) | 112.29 (92.29, 132.78) | 0.68 |
| Monocyte count (X 10$^3$ cell/µL) median (IQR), (213 aspirin arm, 207 placebo arm) | | 0.49 (0.37, 0.63) | 0.46 (0.35, 0.60) | 0.20 |
| Lymphocyte count (X 10$^3$ cell/µL) median (IQR), (213 aspirin arm, 207 placebo arm) | | 1.48 (1.11, 2.01) | 1.52 (1.05, 2.01) | 0.62 |
| Haemoglobin concentration (g/dL) median (IQR) (213 aspirin arm, 209 placebo arm) | | 11.4 (9.95, 13.05) | 11.4 (9.55, 13.00) | 0.92 |
| Platelet count (x 10$^3$ cell/µL) median (IQR), (213 aspirin arm, 209 placebo arm) | | 232 (175, 296) | 219 (161, 274) | 0.07 |
| Soluble CD14 (x 10$^6$ pg/mL) median (IQR), (213 aspirin arm, 211 placebo arm) | | 5.73 (4.21, 8.35) | 6.08 (4.06, 9.25) | 0.65 |
| Soluble P- selectin (x 10$^5$ pg/mL) median (IQR), (213 aspirin arm, 211 placebo arm) | | 1.13 (0.62, 1.90) | 1.16 (0.65, 1.89) | 0.69 |

*(Continued)*

**Table 1.** (Continued)

| Variable | Aspirin arm<br>n = 216 | Placebo arm<br>n = 214 | p- value* |
|---|---|---|---|
| CD4$^+$CD69$^+$ (%) median (IQR), (204 aspirin arm, 202 placebo arm) | 3.69 (1.68, 7.59) | 3.84 (2.00, 7.28) | 0.49 |
| CD8$^+$CD69$^+$ (%) median (IQR), (209 aspirin arm, 202 placebo arm) | 2.33 (1.10, 4.09) | 2.78 (1.47, 5.03) | 0.04 |
| CD4$^+$HLA-DR$^+$CD38$^+$ (%) median (IQR), (206 aspirin arm, 203 placebo arm) | 2.18 (0.97, 3.91) | 1.91 (0.97, 3.75) | 0.55 |
| CD8$^+$HLA-DR$^+$CD38$^+$ (%) median (IQR), (209 aspirin arm, 207 placebo arm) | 2.71 (1.40, 4.83) | 2.72 (1.46, 4.55) | 0.90 |
| CD4$^+$PD-1$^+$ (%) median (IQR), (212 aspirin arm, 208 placebo arm) | 31.05 (22.05, 40.00) | 30.85 (22.15, 41.50) | 0.79 |
| CD8$^+$PD-1$^+$ (%) median (IQR), (212 aspirin arm, 208 placebo arm) | 36.30 (25.90, 47.18) | 37.00 (25.95, 48.20) | 0.94 |

Abbreviations: IQR = interquartile range; eGFR = Estimated glomerular filtration rate.

Notes: *p- values are based on $\chi^2$ analyses for categorical variables and Mann-Whitney U tests for numerical variables.

**Table 2. Participants with virologic suppression or failure at week 24.**

| | Aspirin arm, N = 102<br>n (%) | | | Placebo arm, N = 119<br>n (%) | | | p**, interaction |
|---|---|---|---|---|---|---|---|
| Virologic status | Week 0 | Week 24 | *p** | Week 0 | Week 24 | *p** | Week 24 |
| < 50 RNA copies/mL | 7 (6.86) | 80 (78.43) | < 0.001 | 12 (10.04) | 96 (80.67) | < 0.001 | 0.53 |
| ≥ 1000 RNA copies/mL | 92 (90.20) | 3 (2.94) | < 0.001 | 103 (86.55) | 5 (4.20) | < 0.001 | 0.95 |

Notes: *p-values are based on McNemar test, **p-values are based on interaction (treatment x time) from logistic regression model using generalised estimating equations.

Among participants who made at least one follow-up visit since enrolment, AEs occurring after initiation of study treatment were reported by 146/151 (96.7%) of participants in the aspirin arm and 150/153 (98.0%) in the placebo arm (p = 0.48). No statistically significant differences existed between treatment arms in mild, moderate or severe AEs. The most commonly experienced AE was high BP (26.5% in the aspirin arm and 34.0% in the placebo arm, p = 0.16). Headache was experienced more by participants in the aspirin arm, while low BP was experienced more by participants in the placebo arm (p < 0.05), Table 4.

For all laboratory values (platelet count, haemoglobin concentration, AST and ALT levels) and eGFR, determined at baseline and week 24, median changes from baseline were not significantly different between participants in treatment arms. Among participants who were tested and had normal laboratory values at baseline, the development of thrombocytopenia, anaemia, elevated liver enzymes (AST and ALT) and CKD were not different between the treatment arms at week 24, S3-S8 Tables.

Ten and seven participants died in the aspirin and placebo arms, respectively, during the follow-up period. There were no significant differences in cumulative incidence of death among participants in the aspirin arm (4.63 per 100 person-weeks) and the placebo arm (3.27 per 100 person-weeks), p = 0.47. The rate of loss to follow-up among participants in the aspirin arm (43.62 per 100 person-weeks) was not different from that of participants in the placebo arm (35.58 per 100 person-weeks), p = 0.08, Fig 3.

## Discussion

The present study investigated the effect of low-dose aspirin on virologic, immunologic and clinical responses among PLHIV initiated on ART. To our knowledge, this is the first double-blinded, placebo-controlled randomised trial investigating

**Table 3. Median changes in CD4 count, platelet activation, immune activation, and exhaustion markers from baseline through week 12 and week 24.**

| Markers | Aspirin arm; median changes (IQR) | | | | Placebo arm; median changes (IQR) | | | | p**, interaction | |
| --- | --- | --- | --- | --- | --- | --- | --- | --- | --- | --- |
| | Week 0 – Week 12 | p* | Week 0 – Week 24 | p* | Week 0 – Week 12 | p* | Week 0 – Week 24 | p* | Week 12 | Week 24 |
| CD4 count (cells/mm³) | 132 (51, 212) | < 0.001 | 155 (67, 247) | < 0.001 | 149 (44, 262) | < 0.001 | 157 (92, 277) | < 0.001 | 0.92 | 0.80 |
| CD4 + CD69+ (%) | 0.18 (−1.84, 1.63) | 0.69 | −0.34 (−3.05, 2.10) | 0.36 | −0.58 (−2.63, 1.31) | 0.03 | −0.27 (−2.83, 1.63) | 0.20 | 0.10 | 0.61 |
| CD4 + PD-1+ (%) | 3.00 (−4.60, 10.85) | 0.003 | 0.00 (−9.35, 7.65) | 0.79 | 1.40 (−6.03, 7.70) | 0.27 | −1.80 (−10.80, 4.50) | 0.06 | 0.83 | 0.82 |
| CD4 + HLA-DR + CD38+ (%) | −0.37 (−1.54, 0.29) | < 0.001 | −0.85 (−2.40, −0.08) | < 0.001 | −0.55 (−1.62, 0.31) | < 0.001 | −0.88 (−1.88, −0.12) | < 0.001 | 0.86 | 0.82 |
| CD8 + CD69+ (%) | −0.06 (−1.30, 0.90) | 0.26 | −0.05 (−1.37, 0.80) | 0.35 | −0.42 (−2.07, 0.33) | 0.000 | −0.30 (−1.77, 0.76) | 0.02 | 0.04 | 0.53 |
| CD8 + PD-1+ (%) | −10.00 (−17.00, −0.07) | < 0.001 | −13.20 (−22.05, −4.80) | < 0.001 | −8.15 (−15.73, −0.07) | < 0.001 | −10.70 (−20.60, −3.60) | < 0.001 | 0.44 | 0.53 |
| CD8 + HLA-DR + CD38+ (%) | −1.21 (−3.50, −0.23) | < 0.001 | −1.66 (−4.02, −0.27) | < 0.001 | −1.16 (−2.68, −0.14) | < 0.001 | −1.52 (−2.88, −0.59) | < 0.001 | 0.34 | 0.73 |
| Soluble CD14 (pg/nL) | −0.03 (−2.61, 2.42) | 0.86 | 0.13 (−3.12, 2.32) | 0.81 | −0.16 (−2.65, 2.46) | 0.49 | −0.61 (−3.06, 1.92) | 0.11 | 0.28 | 0.73 |
| Soluble P-selectin (pg/nL) | 0.01 (−0.08, 0.08) | 0.83 | −0.01 (−0.08, 0.05) | 0.21 | 0.01 (−0.06, 0.06) | 0.56 | −0.01 (−0.09, 0.05) | 0.20 | 0.80 | 0.49 |

Abbreviations: IQR = interquartile range.

Notes: *p-values are based on Wilcoxon signed-rank test, **p-values are based on interaction (treatment x time) from linear regression model using generalised estimating equations.

the effect of aspirin on both HIV-specific markers (HVL and CD4 count) and markers of PA, IA and exhaustion among PLHIV in Africa and elsewhere.

We found that the proportions of participants attaining virologic suppression at week 24 were comparable for both participants' groups; those initiated on the low-dose aspirin and ART and those initiated on ART alone. This finding suggests that low-dose aspirin does not improve the effect of ART on HVL. Although a few small studies previously reported the virologic benefits of high-dose aspirin in combination with other drugs and/or micronutrients, these were conducted in a different population from that studied in the present study (ART-naïve PLHIV asymptomatic and those with AIDS not initiating ART) [33,34]. In the current study, almost all participants were initiated on the DTG-based regimen, which is reported to have high efficacy in controlling HVL compared to other regimens such as EFV-based regimens [38–42]. Thus, we presume DTG might have masked any aspirin effect on virologic control if any was present.

In the present clinical trial, both the low-dose aspirin and ART and ART alone treatment arms were associated with increases in CD4 count that were not different in the two arms through week 12 and week 24. This is contrary to previous studies that indicated that treatment containing aspirin resulted in increases in CD4 count [33–35]. The mechanism behind this observation is thought to be the ability of aspirin to inhibit the activation of nuclear factor kappa-B (NF-kB) [43,44]. NF-kB is a transcriptional factor essential for the induction of expression of many cellular and viral genes for infection and inflammation, including those for cytokines, which HIV and other viruses are known to activate [43,45]. However, more effective ART regimens are being developed, and it is unlikely that aspirin may have room to significantly add to HVL lowering or CD4 count increase in PLHIV initiating or already on ART.

In our trial, we observed no significant effect of aspirin on almost all assessed markers of PA, IA or exhaustion. Only CD8 + CD69 +, a marker of acute CD8 + T-cell activation, had a significantly larger decrease in the placebo arm at week 12

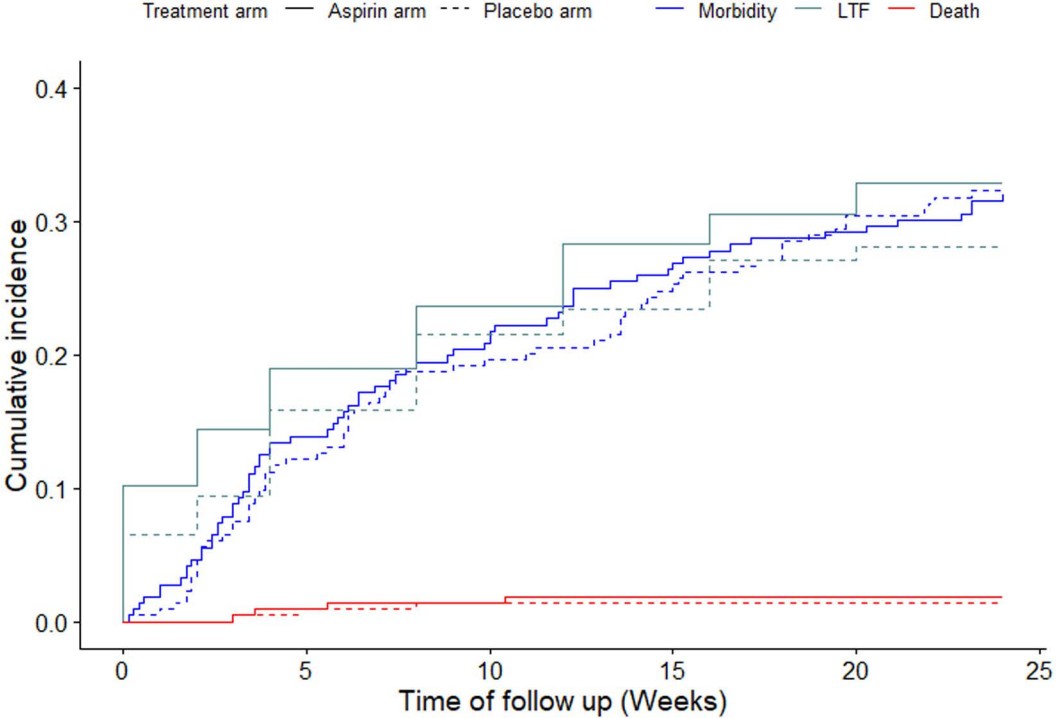

**Fig 2. Cumulative incidence of morbidity, loss to follow-up (LTF) and mortality among study participants by treatment arms from baseline through week 24.**

**Table 4. Most frequently experienced adverse events among study participants by treatment arms from baseline through week 24.**

| Adverse Event | Aspirin arm (N = 151) | | Placebo arm (N = 153) | | p- value* |
|---|---|---|---|---|---|
| | n | % | n | % | |
| UTI | 17 | 11.3 | 15 | 9.8 | 0.67 |
| Tachycardia | 30 | 19.9 | 28 | 18.3 | 0.72 |
| [a]High BP | 40 | 26.5 | 52 | 34.0 | 0.16 |
| Malaria | 14 | 9.3 | 18 | 11.8 | 0.48 |
| Headache | 23 | 15.2 | 12 | 7.8 | 0.04 |
| Cough | 10 | 6.6 | 12 | 7.8 | 0.69 |
| URTI | 14 | 9.3 | 17 | 11.1 | 0.60 |
| Body weakness | 12 | 7.9 | 7 | 4.6 | 0.24 |
| Anaemia | 13 | 8.6 | 9 | 5.9 | 0.36 |
| Peripheral neuropathy | 9 | 6.0 | 8 | 5.2 | 0.76 |
| Nausea/ Vomiting | 17 | 11.3 | 9 | 5.9 | 0.09 |
| [b]Low BP | 8 | 5.3 | 18 | 11.8 | 0.04 |
| Signs of bleeding** | 2 | 1.3 | 5 | 3.3 | 0.25 |
| PUD/Epigastric pain | 4 | 2.6 | 7 | 4.6 | 0.35 |
| Asthma/asthma-like attack | 2 | 1.3 | 0 | 0.0 | 0.16 |
| Tinnitus | 2 | 1.3 | 2 | 1.3 | 1.00 |

Abbreviations: UTI = urinary tract infection; BP = blood pressure; URTI = upper respiratory tract infection; PUD = peptic ulcer disease.

Notes: *p- values are based on $\chi^2$ analyses; [a] systolic BP ≥ 140 mmHg and/or diastolic BP ≥ 90 mmHg; [b] systolic BP < 90 mmHg and/or diastolic BP < 60 mmHg; **signs included bleeding PUD, melaena, haematoma, bleeding haemorrhoids, petechiae, nose bleeding, prolonged per vaginal bleeding.

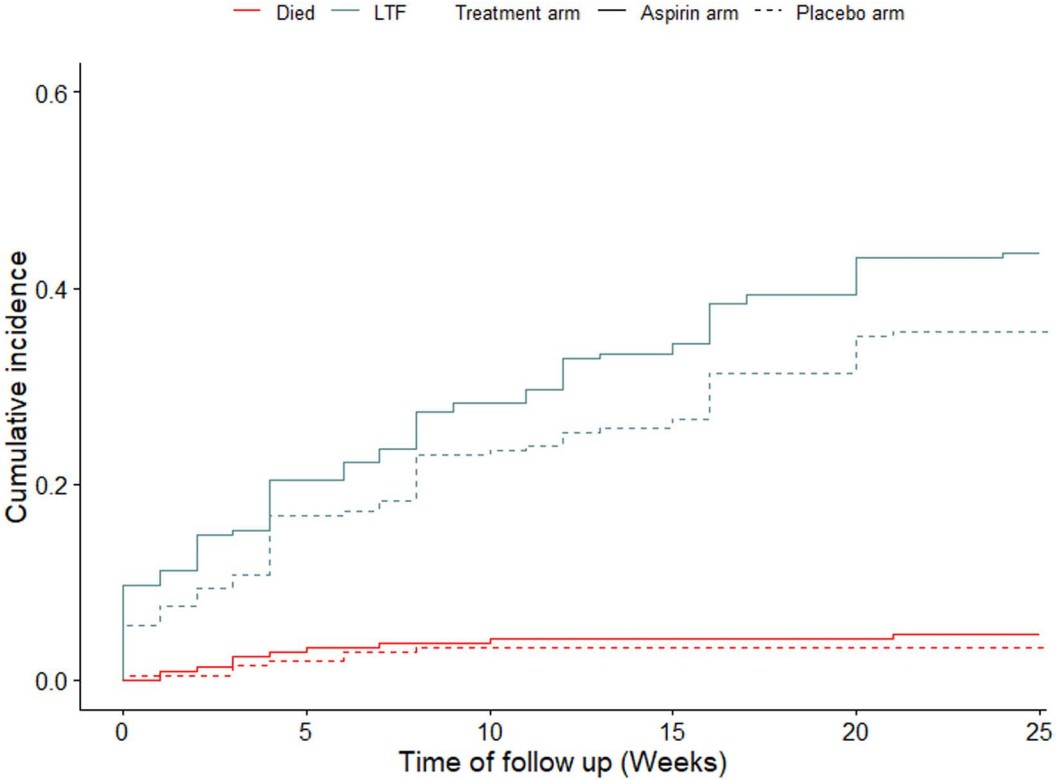

**Fig 3. Cumulative incidence of mortality and loss to follow-up (LTF) among study participants by treatment arms from baseline through week 24.**

but not at week 24 compared to the aspirin arm. However, this finding may be by chance because, at baseline, this marker was significantly higher in the placebo arm compared to the aspirin arm. In one study conducted among virologically suppressed PLHIV on ART and HIV-uninfected controls, treated for a week with 81 mg aspirin daily, a significant decrease from baseline in soluble P-selectin (platelet activation) and T-cell activation was observed compared to HIV-uninfected individuals [7]. Soluble CD14 (sCD14), a marker of monocyte activation, decreased significantly in both the HIV-infected and the HIV-uninfected participants after a week of 81 mg aspirin daily [7]. However, in another study conducted in ART-virologically suppressed PLHIV for a much longer duration (12 weeks), two different doses of aspirin did not affect sCD14 as well as T-cell activation, and exhaustion compared to placebo [32].

The reasons for the difference between our findings and those reported in previous studies are not clear. Although participants in the previous studies had different socio-demographic and clinical characteristics, and the duration of treatment with aspirin was different, these factors may not explain the conflicting findings. Participants in these studies were relatively older, predominantly male and white, had higher CD4 counts and were ART-virologically suppressed. The previous studies have shown no association between these factors and the effect of aspirin on IA [7,32]. In addition, these studies had smaller sample sizes compared.

We observed that low-dose aspirin given together with ART was safe and tolerable. There were no significant differences in morbidity, mortality and abnormal laboratory parameters between the treatment arms. Similar findings have been reported in other studies where low-dose aspirin was administered in PLHIV on ART, including integrase inhibitor-containing ART regimens [32,46]. Should there be any indications or therapeutic benefits, low-dose aspirin may be administered to PLHIV on ART.

Our study had limitations, including a high loss to follow-up contributed by the COVID-19 pandemic situation. We also failed to fully assess the effect of aspirin on virologic response at earlier points (week 8 and week 12). We did not involve health controls that would clarify if levels of markers of PA, IA, and exhaustion observed in our participants were elevated or normal. Nevertheless, our study gives insight into the effect of aspirin on HIV-specific parameters as well as PA, IA and exhaustion in PLHIV.

## Conclusions

Low-dose aspirin given at ART initiation for 24 weeks does not appear to have an additional effect on PA, IA and exhaustion compared to ART alone. Further studies are needed to examine other potential therapeutic targets to control IA and exhaustion in PLHIV as adjuncts to ART.

## Supporting information

**S1 File. Prohibited drugs during the study (Also used for eligibility check).**
(DOCX)

**S1 Table. Logistic regression model using generalised estimating equations on the effects of the treatment arm on virologic response.**
(DOCX)

**S1 Fig. Changes in baseline CD4 count through week 12 and week 24 according to the linear regression model.**
Note: to allow visualisation of all the values, the columns for the arms are slightly offset.
(TIF)

**S2 Fig. Changes in baseline T-cell activation markers through week 12 and week 24 according to the linear regression model. a. Changes in CD4$^+$CD69$^+$ % b. Changes in CD4$^+$HLA-DR$^+$CD38$^+$ % c. Changes in CD8$^+$CD69$^+$ % d. Changes in CD8$^+$HLA-DR$^+$CD38$^+$ %.** Note: to allow visualisation of all the values, the columns for the arms are slightly offset.
(TIF)

**S3 Fig. Changes in baseline T-cell exhaustion, monocyte activation and platelet activation markers through week 12 and week 24 according to the linear regression model. a. Changes in CD4$^+$PD-1$^+$ % b. Changes in CD8$^+$PD-1$^+$ % c. Changes in soluble CD14 (pg/mL) d. Changes in soluble P-selectin (pg/mL).** Note: to allow visualisation of all the values, the columns for the arms are slightly offset.
(TIF)

**S2 Table. Linear regression model using generalised estimating equations on the effects of the treatment arm on CD4 count, platelet activation, immune activation and exhaustion markers.**
(DOCX)

**S3 Table. Laboratory values median changes from baseline among participants at week 24.**
(DOCX)

**S4 Table. Proportion of thrombocytopenia at week 24.** Notes: p-value based on Fisher exact test; thrombocytopenia = platelet count < 150 x 10$^3$ cell/µL.
(DOCX)

**S5 Table. Proportion of anaemia at week 24.** Notes: p-value based on Fisher exact test; anaemia = Haemoglobin concentration < 12.0 g/dl for females and Haemoglobin concentration < 13.0 g/dl for males.
(DOCX)

**S6 Table. Proportion of elevated ALT levels at week 24.** Notes: p-value based on Fisher exact test; elevated ALT levels = ALT level ≥ 1.25 x 34 IU/L for females and ALT level ≥ 1.25 x 45 IU/L for males.
(DOCX)

**S7 Table. Proportion of elevated AST levels at week 24.** Notes: p-value based on Fisher exact test; elevated AST levels = AST level ≥ 1.25 x 31 IU/L for females and AST level ≥ 1.25 x 35 IU/L for males.
(DOCX)

**S8 Table. Proportion of chronic kidney disease at week 24.** Abbreviation: CKD = chronic kidney disease; eGFR = estimated glomerular filtration rate Notes: p-value based on Fisher exact test; CKD = eGFR < 90 mL/min/1.73m$^2$.
(DOCX)

**S2 File. CONSORT 2010 checklist of information to include when reporting a randomised trial.**
(DOC)

**S3 File. Trial study protocol.**
(PDF)

## Acknowledgments

We extend our gratitude to the Temeke District through the District Executive Director, Dar es Salaam, as well as the management of Mwananyamala and Temeke Regional Referral Hospitals and Mbagala Rangi Tatu Hospital for permitting us to conduct this study. We further thank all trial participants, the staff of the participating Care and Treatment Clinics (CTCs) and laboratories. We send our special thanks to Ms. Ellen Hertzmark and Mr. Allen Mulaki for their invaluable support with randomisation and blinding.

## Author contributions

**Conceptualization:** Tosi M. Mwakyandile, Grace A. Shayo, Philip G. Sasi, Ferdinand M. Mugusi, Eligius F. Lyamuya.

**Data curation:** Peter P. Kunambi, Tosi M. Mwakyandile.

**Formal analysis:** Peter P. Kunambi, Tosi M. Mwakyandile.

**Funding acquisition:** Tosi M. Mwakyandile, Ferdinand M. Mugusi, Takamasa Ueno.

**Investigation:** Tosi M. Mwakyandile, Grace A. Shayo, Philip G. Sasi, Eligius F. Lyamuya.

**Methodology:** Tosi M. Mwakyandile, Grace A. Shayo, Philip G. Sasi, Ferdinand M. Mugusi, Godfrey Barabona, Takamasa Ueno, Eligius F. Lyamuya.

**Project administration:** Tosi M. Mwakyandile, Grace A. Shayo, Philip G. Sasi, Eligius F. Lyamuya.

**Resources:** Grace A. Shayo, Godfrey Barabona, Takamasa Ueno.

**Software:** Peter P. Kunambi, Godfrey Barabona, Takamasa Ueno.

**Supervision:** Grace A. Shayo, Philip G. Sasi, Takamasa Ueno, Eligius F. Lyamuya.

**Visualization:** Peter P. Kunambi, Tosi M. Mwakyandile.

**Writing – original draft:** Tosi M. Mwakyandile.

**Writing – review & editing:** Grace A. Shayo, Philip G. Sasi, Peter P. Kunambi, Ferdinand M. Mugusi, Godfrey Barabona, Takamasa Ueno, Eligius F. Lyamuya.

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
