## [Decision Letter · Decision Letter 0]

22 Jul 2025

PONE-D-25-05215Low-dose aspirin is not effective as an adjunct treatment for HIV infection among people living with HIV on dolutegravir-based antiretroviral therapy: a randomised double-blind, parallel-group placebo-controlled trialPLOS ONE

Dear Dr. Mwakyandile,

Thank you for submitting your manuscript to PLOS ONE. After careful consideration, we feel that it has merit but does not fully meet PLOS ONE’s publication criteria as it currently stands. Therefore, we invite you to submit a revised version of the manuscript that addresses the points raised during the review process.

We look forward to receiving your revised manuscript.

Kind regards,

Ismaheel Lawal, MD,

Academic Editor

PLOS ONE

2. We note that the original protocol that you have uploaded as a Supporting Information file contains an institutional logo. As this logo is likely copyrighted, we ask that you please remove it from this file and upload an updated version upon resubmission.

Reviewers' comments:

Reviewer's Responses to Questions

**Comments to the Author**

1. Is the manuscript technically sound, and do the data support the conclusions?

Reviewer #1: Partly

Reviewer #2: Yes

2. Has the statistical analysis been performed appropriately and rigorously? 

Reviewer #1: Yes

Reviewer #2: Yes

3. Have the authors made all data underlying the findings in their manuscript fully available?

Reviewer #1: Yes

Reviewer #2: Yes

4. Is the manuscript presented in an intelligible fashion and written in standard English?

Reviewer #1: Yes

Reviewer #2: Yes

5. Review Comments to the Author

Reviewer #1: The design of the study was a simple randomized parallel strategy with adequate sample size and power utilized.

The data analysis was routine for the objectives. A logistic regression model using generalized estimating equations (GEE) was applied to assess the differences between the treatment arms in proportions of participants with virologic suppression or failure at baseline and 24 weeks. McNemar test was used to compare the proportions of participants in the treatment arms with HVL < 50 and HVL ≥ 1,000 RNA copies/mL at baseline and 24 weeks. A Linear regression model using GEE was used to analyze differences in CD4 count, markers of platelet activation, immune activation, and exhaustion between the arms at baseline, 12 and 24 weeks. In both types of analyses, the interaction of time and treatment arm represented the treatment effect.

The overall conclusion that Low-dose aspirin initiated with ART through 24 weeks did not impact virologic or immunologic markers among PLHIV was supported by the analysis performed. On a minor note, the detail results of the logistic regression could have been enhanced. What exactly were the variables considered in the model and were there any statistically significant confounders or possible confounders? One suspects not. However, some clarification would help. Also, was a visit or admission to a health facility for medical attention/care due to illness the only morbidity?

The authors have , to their credit, noted the high lost to follow up as a study limitation. Perhaps future strategies should focus on recruitment retention given the short duration of the study.

Reviewer #2: Overall, the study has a robust design with clearly defined outcomes and transparency. The attrition seems to be balanced in both cohorts. Although all time points for primary outcome could not be achieved, 24-month analysis for primary outcome is acceptable in view of longitudinal nature of this disease, as well as employed equivalent duration of aspirin treatment.

Apart from minor corrections in verbiage that authors should re-read and make, I do not have notable comments.

6. PLOS authors have the option to publish the peer review history of their article (what does this mean? ). If published, this will include your full peer review and any attached files.

**Do you want your identity to be public for this peer review?** For information about this choice, including consent withdrawal, please see our Privacy Policy .

Reviewer #1: No

Reviewer #2: No

---

## [Author Response · Author response to Decision Letter 1]

28 Jul 2025

The responses to specific reviewer and editor comments are detailed in the attached file labeled 'Response to reviewers'

---

## [Decision Letter · Decision Letter 1]

12 Aug 2025

Low-dose aspirin is not effective as an adjunct treatment for HIV infection among people living with HIV on dolutegravir-based antiretroviral therapy: a randomised double-blind, parallel-group placebo-controlled trial

PONE-D-25-05215R1

Dear Dr. Mwakyandile,

We’re pleased to inform you that your manuscript has been judged scientifically suitable for publication and will be formally accepted for publication once it meets all outstanding technical requirements.

Kind regards,

Ismaheel Lawal, MD,

Academic Editor

PLOS ONE

Additional Editor Comments (optional):

Reviewers' comments:

Reviewer's Responses to Questions

**Comments to the Author**

1. If the authors have adequately addressed your comments raised in a previous round of review and you feel that this manuscript is now acceptable for publication, you may indicate that here to bypass the “Comments to the Author” section, enter your conflict of interest statement in the “Confidential to Editor” section, and submit your "Accept" recommendation.

Reviewer #1: All comments have been addressed

2. Is the manuscript technically sound, and do the data support the conclusions?

Reviewer #1: (No Response)

3. Has the statistical analysis been performed appropriately and rigorously? 

Reviewer #1: (No Response)

4. Have the authors made all data underlying the findings in their manuscript fully available?

Reviewer #1: (No Response)

5. Is the manuscript presented in an intelligible fashion and written in standard English?

Reviewer #1: (No Response)

6. Review Comments to the Author

Reviewer #1: (No Response)

7. PLOS authors have the option to publish the peer review history of their article (what does this mean? ). If published, this will include your full peer review and any attached files.

**Do you want your identity to be public for this peer review?** For information about this choice, including consent withdrawal, please see our Privacy Policy .

Reviewer #1: No

---

## [Editor Report · Acceptance letter]

PONE-D-25-05215R1

PLOS ONE

Dear Dr. Mwakyandile,

I'm pleased to inform you that your manuscript has been deemed suitable for publication in PLOS ONE. Congratulations! Your manuscript is now being handed over to our production team.

Kind regards,

on behalf of

Dr. Ismaheel Lawal

Academic Editor

PLOS ONE